# Neutrophils and Asthma

**DOI:** 10.3390/diagnostics12051175

**Published:** 2022-05-08

**Authors:** Akira Yamasaki, Ryota Okazaki, Tomoya Harada

**Affiliations:** Department of Multidisciplinary Internal Medicine, Division of Respiratory Medicine and Rheumatology, Faculty of Medicine, Tottori University, Yonago 683-8504, Japan; okazaki0222@tottori-u.ac.jp (R.O.); tomo.h.308@tottori-u.ac.jp (T.H.)

**Keywords:** asthma, biomarkers, biologics, eosinophils, inflammation, neutrophils, treatment

## Abstract

Although eosinophilic inflammation is characteristic of asthma pathogenesis, neutrophilic inflammation is also marked, and eosinophils and neutrophils can coexist in some cases. Based on the proportion of sputum cell differentiation, asthma is classified into eosinophilic asthma, neutrophilic asthma, neutrophilic and eosinophilic asthma, and paucigranulocytic asthma. Classification by bronchoalveolar lavage is also performed. Eosinophilic asthma accounts for most severe asthma cases, but neutrophilic asthma or a mixture of the two types can also present a severe phenotype. Biomarkers for the diagnosis of neutrophilic asthma include sputum neutrophils, blood neutrophils, chitinase-3-like protein, and hydrogen sulfide in sputum and serum. Thymic stromal lymphoprotein (TSLP)/T-helper 17 pathways, bacterial colonization/microbiome, neutrophil extracellular traps, and activation of nucleotide-binding oligomerization domain-like receptor family, pyrin domain-containing 3 pathways are involved in the pathophysiology of neutrophilic asthma and coexistence of obesity, gastroesophageal reflux disease, and habitual cigarette smoking have been associated with its pathogenesis. Thus, targeting neutrophilic asthma is important. Smoking cessation, neutrophil-targeting treatments, and biologics have been tested as treatments for severe asthma, but most clinical studies have not focused on neutrophilic asthma. Phosphodiesterase inhibitors, anti-TSLP antibodies, azithromycin, and anti-cholinergic agents are promising drugs for neutrophilic asthma. However, clinical research targeting neutrophilic inflammation is required to elucidate the optimal treatment.

## 1. Introduction 

Asthma is a common chronic airway disease that affects about 350 million people worldwide and varies in prevalence from country to country. In Japan, the prevalence is 9–10% and the number of patients with asthma was 1,177,000 in 2014 [1,2]. Diagnosis of asthma is based on a history or current symptoms, such as chest tightness, wheezing, dyspnea, and cough, together with variable expiratory airway limitation assessed by peak expiratory flow or spirometry. Chronic airway inflammation is an important feature of asthma and is characterized by the presence of eosinophils, basophils, mast cells, neutrophils, T helper 2 (Th2) cells, type 2 innate lymphoid cells (ILC2), CD8^+^ T cells, B cells, and dendritic cells [3,4,5]. In the Japanese Guidelines for Adult Asthma, a diagnosis is based on: (I) repetitive symptoms, such as paroxysmal dyspnea, wheezing, chest tightness, and cough; (II) reversible airflow limitation; (III) airway hyper-responsiveness; (IV) airway inflammation; (V) an atopic state; and (VI) exclusion of other cardiopulmonary disease [2].

Asthma is a heterogenous airway disease, and since the 2000s, cluster analyses have identified several phenotypes [6,7,8]. The common phenotypes are allergic asthma; non-allergic asthma; adult-onset (late-onset) asthma; asthma with persistent airflow limitation, and asthma with obesity [9]. The Severe Asthma Research Program (SARP) identified five phenotypes in patients with severe and non-severe asthma [10]. Kuo et al. found three transcriptome-associated clusters (TACs) in patients with asthma. TAC1 is characterized by immune receptors and a sputum eosinophil increase, TAC2 is characterized by interferon-, tumor necrosis factor-, and inflammasome-associated genes and a sputum neutrophil increase, and TAC3 is characterized by genes associated with metabolic pathways, ubiquitination, and mitochondrial function, with no sputum increase [11]. 

Neutrophils are the most abundant cells in peripheral blood and are stored in pulmonary capillary beds [12]. These cells play important roles in the innate immune system by killing microbes, phagocytosis, granule release, and formation of neutrophil extracellular traps (NETs). The role of neutrophils in asthma has been studied, but there is much debate about the presence of neutrophilic asthma [13,14,15,16]. Since glucocorticoids enhance the survival of neutrophils, which constitutively express glucocorticoid receptor β (GRβ) [17,18], the elevation of neutrophil levels in the asthmatic airway is thought to be a consequence of corticosteroid treatment. However, neutrophils are also observed in steroid-naïve patients with asthma [19,20,21,22] and several studies have found evidence that neutrophilic inflammation is associated with severe asthma and with asthma exacerbation [23,24]. A cluster analysis has shown that sputum neutrophil counts were associated with more severe phenotypes [25]. Recently, Minchem et al. reviewed the pathology of chronic lung diseases, including asthma [26]. They described the heterogeneity of neutrophils and their interactions with several immune and structural cells, identifying anti-inflammatory, pro-resolving, and pro-repair functions via direct cell-to-cell communication as well as via soluble mediators [26]. Neutrophils also connect with other cells via exosomes and extracellular vesicles [27]. In chronic lung diseases, an overabundance of neutrophils may exacerbate inflammation and remodeling [26]. Therefore, neutrophilic inflammation is involved in the heterogeneity of asthma, and neutrophil-targeted treatment may be important for severe asthma. The pathogenesis, definition, and biomarkers of neutrophilic asthma and potential therapy for neutrophilic asthma are discussed in this review. 

## 2. Definition of Neutrophilic Asthma

The phenotype of asthma is generally categorized by the cell profile of induced sputum. In a healthy person, this profile has 0.4 ± 0.9% eosinophils and 37.5 ± 20.5% neutrophils, with means plus 2SD and 90th percentiles of 2.2% and 1.1% for eosinophils, and 77.7% and 64.4% for neutrophils, respectively [28]. Eosinophilic asthma is defined as an increase in eosinophils to above 2% or 3% and neutrophilic asthma as an increase in neutrophils to above 60% or 76% in induced sputum [29]. Paucigranulocytic asthma is defined as neutrophils < 76% and eosinophils < 3%, and conversely, mixed granulocytic asthma is defined as neutrophils > 76% and eosinophils > 3% [30]. However, there is still no clear definition of neutrophilic asthma [13]. In children, neutrophil-predominant severe asthma is defined using a cut-off of ≥5% neutrophils in bronchial lavage fluid [31]. Alternative methods, such as nasal wash or nasal lavage, have also been used to evaluate neutrophilic asthma or non-eosinophilic asthma [32]. 

## 3. Association of Eosinophils and Neutrophils

Coexistence of neutrophils and eosinophils occurs in severe asthma [10,33,34], and recent studies have shown that patients with asthma with a mixture of neutrophilic and eosinophilic inflammation had accelerated decline of respiratory function [35,36,37]. In studies of the coexistence mechanism, Nagata et al. found that activation of neutrophils may induce migration of eosinophils through the basement membrane via interleukin-8 (IL-8) [38], and that leukotriene B4 (LTB4)-activated neutrophils which induced eosinophil migration and Toll-like receptor 4 (TLR4) expression on neutrophils may be involved in this mechanism [36,39]. Theophylline attenuates trans-basement membrane migration of eosinophils in vitro by suppressing superoxide anion generation [40]. Lavinskinene et al. showed that the sputum neutrophil counts after bronchial allergen challenge were related to peripheral blood neutrophil chemotaxis in patients with asthma [41].

## 4. Pathogenesis of Asthma

### 4.1. TSLP

Thymic stromal lymphopoietin (TSLP) is secreted from a variety of cells, including basophils, mast cells, and airway epithelial cells [42]. In the human airway, airway epithelial cells secrete TSLP by recognition of allergens, viruses, pollutants and cigarette smoke, bacteria, and other external stimuli by pattern recognition receptors (PRPs) [43]. TSLP triggers allergic/eosinophilic and non-allergic/eosinophilic inflammation [44,45], and is also involved in neutrophilic inflammation in asthma. TSLP and TLR3 ligands promote conversion of naïve T cells to Th17 cells [46] and subsequently induce neutrophil recruitment via IL-8 and GM-CSF from airway epithelial cells [47]. TSLP polymorphism may also be related to allergic disease and eosinophilia in patients with asthma [48]. 

### 4.2. IL-17

IL-17 is a key cytokine in neutrophilic asthma. IL-17 and IL-17A are produced by Th17 cells and ILC3 cells, and may stimulate epithelial cells and fibroblasts and induce neutrophil activation and migration via IL-6, IL-8, and tumor necrosis factor-α (TNF-α). IL-17 induces glucocorticoid receptor (GR) β on epithelial cells in patients with asthma [49]. This may be related to glucocorticoid insensitivity in neutrophilic asthma. IL-17 induces eotaxin expression in human airway smooth muscle (HASM) cells [50], which may be linked to mixed neutrophilic and eosinophilic inflammation in asthma. IL-17 is increased in bronchial biopsy in severe asthma [51] and in sputum from patients with moderate-to-severe asthma [52]. Bulles et al. showed that the *IL17* mRNA level correlated with the *IL8* mRNA level and with CD3 gamma cell and neutrophil counts, which suggested a link between IL-17 and neutrophilic inflammation [52]. IL-17 also enhances IL-1β-mediated IL-8 release from HASM cells [53], and the IL-17/Th17 axis is involved in microbiomes in the development of asthma [54]. 

### 4.3. Bacterial Colonization and Microbiome in the Airway in Neutrophilic Asthma

The intestinal and respiratory microbiomes are both thought to be associated with the pathogenesis of asthma [55]. In patients with neutrophilic asthma, 50% of patients have bacterial infection based on bronchoalveolar lavage [56], and at the time of asthma exacerbation, 87.8% of patients have bacteria in sputum, with neutrophils > 65% [13]. Recent studies have shown that bacterial microbiome profiles in the airway were associated with neutrophil inflammation in asthma [57,58,59] and that the Th17/IL-17 axis was involved in this process [60,61]. Microbiome-derived cluster analysis of sputum in severe asthma showed two distinct phenotypes: cluster 1 had less-severe asthma and commensal bacterial profile, and higher bacterial richness and diversity; cluster 2 had more severe asthma with a reduced commensal bacterial profile, clear deficiency of several bacterial species, and neutrophilic inflammation [57]. The intestinal microbiome has also been linked to the development of asthma, but its relationship with neutrophilic inflammation in asthma is unclear [62]. 

### 4.4. Obesity

Obesity increases the risk of asthma development [63,64,65,66], worsens asthma control and severity [8,67], increases hospitalization [68], and reduces responses to inhaled corticosteroids (ICS) alone or in conjunction with a long-acting β2 agonist (LABA) [68,69,70]. In cluster analyses, obesity-related asthma has been grouped into non-Th2 asthma, with later onset, female preponderance, and severe symptoms [7,8,10]. Obesity is associated with inflammatory adipokines including leptin, resistin, lipocain 2, IL-6, TNF-α, IL-1β, and IFN-γ [71,72,73,74,75]. These mediators induce airway inflammation. In a mouse obese asthma model, ILC3 stimulated by IL-1β, IL-6, or IL-23 produced IL-17A [76]. IL-17A alone or in combination with TNF-α has been shown to induce IL-8 production from epithelial cells [77], and cigarette smoke can also enhance IL-17A-induced IL-8 and IL-6 production [78,79,80,81]. IL-6 and IL-8 recruit and activate neutrophils in an asthmatic airway [41,81]. In obese patients with asthma, IL-17 is associated with steroid resistance by dysregulation of GRα and GRβ [82], while in human bronchial epithelial cells, IL-17A induces glucocorticoid insensitivity [83]. Insulin resistance and vitamin D deficiency related to obesity may aggravate airway remodeling and hyper-responsiveness by enhancing leptin, transforming growth factor (TGF)-β1, IL-1β, and IL-6 expression [84,85,86,87], which might then promote neutrophilic inflammation. 

### 4.5. NETs and NETosis

Neutrophil extracellular traps (NETs) were first described by Brinkmann et al. [88]. Neutrophils stimulated by bacteria or inflammatory mediators, such as IL-8, platelet activating factor, and lipopolysaccharide (LPS), release NETs that include neutrophil elastase, cathepsin G, myeloperoxidase, defensins, lactoferrin, histones, pentraxin 3, reactive oxygen species (ROS), and DNA to captivate and kill bacteria [89]. NETosis is an active form of neutrophil death related to NETs formation [88]. Several studies have related NETs to the pathogenesis of autoimmune disease, cancer, and atherosclerosis [90,91]; dysregulation of NETs may also result in asthma pathobiology, although the mechanisms associated with NETs are not fully understood. In a mouse model, allergen exposure with endotoxin induced NETosis [92]. In severe neutrophilic asthma, Krishnamoorthy et al. determined that cytoplasts and neutrophils positively correlated with IL-17 levels in the bronchoalveolar fluid [92]. The sputum extracellular DNA (eDNA) level has been correlated with expressions of IL-8, IL-1β, and NLRP3 [93], and Lachowicz-Scroggins et al. found that high extracellular DNA (eDNA) in sputum was associated with poor asthma control, mucus hypersecretion, and oral steroid use in patients with asthma [94]. The same group also showed that the eDNA level was correlated with neutrophil inflammation, NET components, caspase-1 activity, and IL-1β. In vitro, epithelial damage caused by NETs has been prevented by DNase [94]. These studies indicate that NETs and eDNA are related to severe neutrophilic asthma.

### 4.6. NLRP3 Inflammasome and Asthma

Nucleotide-binding oligomerization domain-like receptor family pyrin domain-containing (NLRP) inflammasomes are a critical component of the innate immune system and they play an important role in activation of inflammation. NLRP3, an NLR family PRP, responds to pathogen-associated molecule patterns (PAMPs) or danger (damage)-associated molecular patterns (DAMPs). Activation of NLRP3 inflammasomes is mediated by two signals: the priming signal triggered by PAMPs, DAMPs, IL-1β, and TNF-α; the second (activation) signal mediated by extracellular ATP, RNA viruses, particulate matter, ionic flux, ROS, mitochondrial dysfunction, and lysosomal damage. Upon activation of NLRP3 inflammasomes, IL-1β and IL-18 are secreted [95,96]. Dysregulation of NLRP3 inflammasome activation is related to Alzheimer’s disease [97], Parkinson’s disease [98], diabetes mellitus, atherosclerosis [99], and pulmonary inflammatory disorders, including lung fibrosis [100], acute exacerbation of interstitial pneumonia [101], sarcoidosis [102], asbestosis, and silicosis [103]. Since human lungs are exposed to many endogenous and exogenous noxious stimuli, including viruses, bacteria, cigarette smoke, and particulate matter, the innate immune response in the airway via NLRP3 inflammasomes is important. However, excess or persistent activation of NLRP3 inflammasomes by allergens or irritants has been shown to induce persistent inflammation and tissue damage in the airway of patients with asthma [104,105]. In these patients, the sputum NLRP3 level was increased and was correlated with neutrophilic airway inflammation [106,107]. NLRP3 expression has also been shown to be increased in obese patients with asthma [108]. Kim et al. found that a high-fat diet induced airway hyper-reactivity and increased *NLRP3*, *IL17A*, and *IL1B* mRNA in an obese mouse model [76], suggesting that obesity-induced airway hypersensitivity is mediated by NLRP3 inflammasomes that are activated by fatty acids or cholesterol crystals from macrophages in adipose tissue or in the lungs [76]. In other experimental models, NLPR3 and apoptosis-associated speck-like protein containing CARD (ASC)-deficient mice exhibited reduced airway inflammation [109]. Ovalbumin (OVA) mouse models with alum [110], LPS, *Aspergillus fumigatus* [111], *Chlamydia muridarum*, or *Haemophilus influenzae* infection also have been shown to have increased NLRP3 [106]. In this latter model, neutrophil depletion suppressed IL-1β-induced airway hyper-responsiveness. 

### 4.7. S100A8/A9, HMGB-1, RAGE, and TLR4

The S100A8/A9 complex belongs to the Ca^2+^-binding S100 protein family and is a DAMP protein complex expressed in neutrophils, monocytes, and macrophages [112,113]. High mobility group box 1 (HMGB-1), which is also a DAMP protein, a non-histone, chromatin-associated nuclear protein is released from necrotic, inflammatory, macrophage, dendritic, natural killer, and resident cells (epithelial cells, smooth muscle cells, and fibroblasts) [114,115,116,117]. TNF-α, IL-1β, and IFN-γ induce HMBG-1 release from activated macrophages [118,119]. HMBG-1 and S100A8/S100A9 bind to two types of receptors: the receptor for advanced glycation end products (RAGE) and TLR-4. RAGE is expressed on lung [120], skeletal muscle, heart, liver, kidney [121], lung epithelial, and immune cells [122,123,124,125,126]. Perkins et al. showed that knockout of *RAGE* abolished type 2 cytokine-induced airway inflammation and mucus hyperplasia in a mouse model [127]. Oczypok et al. reported that RAGE induced asthma/allergic airway inflammation by promoting IL-33 expression, and that ILC2 accumulation was critical in the pathogenesis of asthma in a mouse model [128]. 

TLR4 is also expressed on B cells [129], T cells [130], monocytes, macrophages [131], and neutrophils [132]. S100A8/A9 and HMGB-1 might be involved in the pathobiology of remodeling in asthma by promoting inflammation and tissue repair in the airway [117]. In a mouse model, blocking HMGB-1 and TLR-4 attenuated disonoyl phthalate-induced asthma [133]. HMGB-1 is increased in OVA-induced asthma [134]. In patients with asthma, the sputum HMGB-1 level is increased and inversely correlated with the percentage predicted forced expiratory volume in 1 s (%FEV1) and FEV1/forced vital capacity (FVC) ratio. The HMGB-1 level is also associated with the severity of asthma and neutrophils in sputum [135,136]. An endogenous form of RAGE (esRAGE), which is a decoy receptor for AGE, was elevated in sputum from a patient with asthma; however, the esRAGE level was not associated with asthma severity [135], in contrast to the RAGE level [136]. Since HMGB-1 stimulates TNF-α, IL-6, and IL-8 production from monocytes [137,138], it might be a key player in inducing neutrophilic asthma. Recent studies have shown that a soluble form of RAGE prevents Th17-mediated neutrophilic asthma by blocking HMBG1/RAGE signaling in a mouse model [139]. In patients with neutrophilic asthma, decreased sRAGE was associated with asthma severity [140], and a recent study showed that sRAGE was associated with low eosinophil count and IgE in children with asthma [141]. RAGE has been linked to cigarette-smoke-induced neutrophilic inflammation and airway hyper-responsiveness in a mouse model, but TLR4, another receptor for HMGB-1 and S100A8/A, was not involved [142]. Furthermore, *AGER* (which encodes RAGE) expression, rather than TLR4 expression, was significantly correlated with the sputum neutrophil count and airway hyper-responsiveness in patients with chronic obstructive pulmonary disease (COPD) [142]. Therefore, HMGB-1 and sRAGE might be biomarkers for neutrophilic asthma.

### 4.8. House Dust Mites and Neutrophilic Asthma 

House dust mites (HDMs) are the most important allergen for the development and worsening of allergic asthma, with 90% of cases of pediatric asthma sensitized to HDMs. Many studies of allergic and eosinophilic asthma have been conducted using a mouse model sensitized to HDMs, and several recent studies have described neutrophilic or mixed-granulocytic asthma models. Menson et al. reported a novel BALB/c female mouse model using *Mycobacterium tuberculosis* extract, complete Freund’s adjuvant, and HDM, in which the bronchial alveolar lavage fluid (BALF) contained 80% neutrophils and 10% eosinophils [143]. Mack et al. described an old (9 months) C57BL/6 female mouse model sensitized to HDMs that showed elevated neutrophils in BALF as compared with young (3 months) mice, as a model of adult-onset asthma [144]. Sadamatsu et al. found that a high-fat diet induced elevated neutrophils in BALF in an HDM-sensitized mouse model [145]. Neutrophil counts in the sputum of patients with chronic neutrophilic asthma have been shown to be correlated with the serum HDM-specific IgG levels, and these patients have HDM-derived enolase in their serum [146]. In the same study, HDM-derived enolase was shown to induce epithelial barrier disintegration and neutrophilic inflammation in a mouse model [146]. Blockade of leukotriene B4 receptor 1 (BLT1)/BLT2 by antagonists can reduce neutrophil infiltration based on findings in an HDM- and LPS-induced mouse asthma model [147]. IL-1β was found to be required to promote neutrophilic inflammation in an HDM-sensitized and viral-exacerbated model, using double-stranded RNA to mimic rhinovirus [148]. In contrast, Patel et al. found neutrophil depletion in an HDM allergic airway disease mouse, with this depletion enhancing Th2 inflammation by inducing G-colony stimulating factor-induced ILC2 activation and cytokine production [149].

### 4.9. Electric, Heat-Not-Burn Cigarettes, and Combustible Cigarettes

Almost one-quarter of patients with adult asthma are thought to have smoking habits. Several studies have also shown that the efficacy of ICS is reduced in patients with asthma who are exposed to smoking [150,151,152]. Passive smoking in a family increases the use of drugs for pediatric asthma [153]. E-cigarette or electric cigarette (vapor) exposure induces neutrophil protease, matrix metalloproteinase-2 (MMP-2), and MMP-9 in healthy subjects [154]. Schweitzer et al. showed that e-cigarette use was independently associated with asthma in adolescents [155]. A study from Korea also showed an association of e-cigarette use with asthma diagnosis and absence from school due to asthma [156]. E-cigarette liquid has been shown to induce IL-6 production from human epithelial cells and addition of nicotine further increased IL-6 production [157], while electronic nicotine delivery systems using aerosols also induced IL-6 and IL-8 secretion [158].

A 2015 internet survey showed that the use of heat-not-burn (HNB) cigarettes among patients with asthma was 0.0% in Japan [159]. The first HNB cigarette, IQOS, was released in 2014 in Japan, and the harmfulness of HNB cigarettes to asthma remains uncertain. However, HNB cigarettes contain nicotine and many other toxins [160,161], as well as particulate matter [162], and thus, may worsen asthma control by inducing neutrophilic inflammation. Further studies are needed to examine how HNB cigarettes affect asthma pathogenesis and neutrophilic inflammation [80]. In patients with mild asthma, combustible cigarette smoking increases neutrophil counts, and IL-17A, IL-6, and IL-8 levels [80]. Exposure of human epithelial cells to cigarette smoke extracts, IL-17A, and aeroallergens has been shown to induce IL-6 and IL-8 production, which may be associated with the neutrophil accumulation in asthmatic airways [80]. In a rat model, the late asthmatic response to OVA increased with cigarette smoke (CS) exposure as compared with no exposure. ICS decreased eosinophil and lymphocyte accumulation with and without CS exposure but did not decrease neutrophil accumulation with CS exposure [163]. Quitting smoking and avoiding environmental smoking can resolve neutrophil inflammation in patients with asthma who smoke. A combination of pharmacotherapy using bupropion and varenicline with counseling was most effective for smoking cessation [164]. Smoking cessation-support therapy using a smartphone application has recently been covered by insurance in Japan [165]. 

### 4.10. Air Pollution

Relationships of air pollution with asthma development or exacerbation have been reported for several years. Examples of outdoor or indoor pollution include diesel exhaust, foreign workplace matter, ozone, nitrogen dioxide, sulfur dioxide, second-hand smoke, heating sources, cooking smoke, and molds [166,167,168]. These pollutants induce asthma exacerbation through oxidative stress and damage, airway remodeling, inflammatory pathways, immunological responses, and enhancement of airway sensitivity [166,168]. Particulate matter induces Th2 and Th17 inflammation in allergic conditions and this induces eosinophilic and neutrophilic inflammation in asthma [169,170,171,172]. In an in vivo study, ozone exposure induced IL-8 secretion from epithelial cells [173], which was related to neutrophil accumulation in the airway after exposure to ozone in patients with asthma [174]. 

### 4.11. Gastroesophageal Reflux Disease

Gastroesophageal reflux disease (GERD) is a common comorbidity in asthma, and the severity of asthma is increased when complicated with GERD [175]. In the SARP study, a subgroup of patients with asthma defined as having a low pH in exhaled breath condensate had a high body mass index (BMI) and high neutrophilic airway inflammation, and had GERD as a complication [176]. GERD is often accompanied by mixed eosinophilic and neutrophilic inflammation (reviewed in [177]). Simpson et al. found that patients with neutrophilic asthma had a high prevalence of rhinosinusitis and symptoms of GERD as compared with patients with eosinophilic asthma [178]. The mechanism through which GERD induces or enhances airway inflammation in asthma has not been determined, but GERD is associated with obesity [179], which may lead to neutrophilic inflammation, as mentioned above. The triangle of inflammation, obesity, and GERD with sleep disordered breathing syndrome is important in children with asthma [180]. 

Figure 1 shows the pathology of neutrophilic asthma (Figure 1).

## 5. Biomarkers of Neutrophilic Asthma

Non-type 2 subtypes of asthma, including neutrophilic and paucigranulocytic asthma, are difficult to diagnose because of a lack of appropriate biomarkers. However, recent studies have suggested promising diagnostic biomarkers for neutrophilic asthma (Table 1). 

### 5.1. YKL40

Chitinase-3-like protein (YKL-40) is a human glycoprotein that is released from several cell types, including neutrophils, macrophages, and epithelial cells. YKL-40 is involved in the pathogenesis of many diseases, including rheumatoid arthritis [192], multiple sclerosis [193], chronic obstructive lung disease [194,195], Alzheimer’s disease [196], and asthma [181,197]. Serum YKL-40 levels are related to asthma severity, while lung YKL-40 levels are correlated with airway remodeling [181,182]. In the multicenter BIOAIR study, the serum YKL-40 level was negatively correlated with lung function (FEV1% predicted, FVC, and FEV1/FVC), but not with fraction of exhaled nitric oxide or blood and sputum eosinophil and neutrophil counts [182]. Cluster analyses have shown that high serum YKL-40 levels were associated with neutrophilic asthma and paucigranulocytic asthma [183] and that patients with high serum YKL-40 had severe airflow obstruction and near fatal or frequent exacerbation [183]. The serum YKL-40 level has been shown to be positively correlated with blood neutrophils, IL-6, and sputum IL-1β [119], while the sputum YKL-40 level has been shown to be strongly correlated with neutrophilic asthma and sputum myeloperoxidase, and was associated with sputum IL-8 and soluble IL-6 receptor levels [187]. Therefore, serum and sputum YKL-40 levels are useful biomarkers for neutrophilic asthma.

### 5.2. Hydrogen Sulfide

Nitric oxide is a biomarker of type 2 inflammation and carbon monoxide is a partial biomarker of asthma severity [198,199]. Hydrogen sulfide (H_2_S) is the third biomarker in breath, and sputum H_2_S is a novel biomarker of neutrophilic asthma. Sputum H_2_S levels are correlated with neutrophils in sputum and airflow limitation [184,185,186], and the sputum-to-serum H_2_S ratio predicts the risk of asthma exacerbation [186]. Therefore, sputum H_2_S is a diagnostic marker for neutrophilic asthma and a predictor of exacerbation when combined with serum H_2_S. These biomarkers are also elevated in asthma-COPD overlap [200].

### 5.3. Myeloperoxidase

Myeloperoxidase (MPO) is a marker of neutrophil activation. Serum MPO has been shown to be elevated in ANCA-associated vasculitis, including microscopic polyangiitis and eosinophilic granulomatous polyangiitis, while sputum MPO has been shown to correlate positively with sputum YKL-40 levels [187] and sputum neutrophils [23]. Thus, sputum MPO is a useful biomarker for neutrophilic asthma, whereas elevation of serum MPO is thought to be a marker for small vessel vasculitis. 

### 5.4. Blood Neutrophil Count

The peripheral blood neutrophil count is not appropriate as a surrogate marker for neutrophilic asthma defined based on sputum cell differentiation [201,202,203]. However, neutrophilia has been shown to be associated with chronic airway obstruction [189] and an annual decline in FEV1 [188]. The sputum neutrophil count after bronchial allergen challenge has been shown to be related to peripheral blood neutrophil chemotaxis in patients with asthma [41].

### 5.5. MicroRNA

Several studies have shown that microRNAs (miRNAs) are biomarkers for asthma. Panganiban et al. found upregulation of miRNA-1248 in patients with asthma [204] and also showed that miRNAs in serum could be used to phenotype asthma [205]. Huang et al. revealed that miR-199a-5p in sputum and plasma was increased in neutrophilic asthma [190] and showed that levels of miRNA-199a-5p secreted from human LPS-stimulated peripheral neutrophils were inversely correlated with FEV1 [190]. A genome-wide analysis of miRNAs in sputum from patients with asthma showed that *hsa*-miR-223-3p was expressed in neutrophils and was associated with neutrophil counts in response to ozone exposure [206]. Maes et al. showed that miR-223-3p, miR-142-3p, and miR-629-3p were upregulated in severe, neutrophilic asthma [191]. Therefore, several miRNAs are biomarkers for diagnosis of neutrophilic asthma, and they are also considered to be therapeutic targets [207,208]. 

## 6. Airway Remodeling in Neutrophilic Asthma

Airway remodeling in asthma is caused by chronic airway inflammation and is a characteristic feature of chronic asthma. The pathological changes in airway remodeling involve mucous metaplasia, thickening of the reticular basement membrane, increases of goblet cells and mucus hypersecretion, shedding of epithelial cells, submucosal infiltration of inflammatory cells, extracellular matrix deposition, airway smooth muscle (ASM) cell hyperplasia, and hypertrophy. Neutrophilic asthma and airway remodeling are not fully understood, but several studies have shown that inflammatory mediators, such as LTB4, IL-6, IL-8, and TNF-α, which are related to neutrophilic inflammation, were elevated in an asthmatic airway. Several of these mediators and cytokines have also been shown to be elevated in neutrophilic asthma, of which LTB4, IL-8, TNF-α, IL-17, and IL-6 may be related to airway remodeling. Figure 2 shows neutrophilic inflammation-associated airway remodeling in asthma (Figure 2).

### 6.1. Leukotriene B4

In severe asthma, leukotriene B4 (LTB4) is increased in sputum, BALF, exhaled breath condensate, urine, and arterial blood [209]. LTB4 is a chemoattractant mediator of neutrophils [210] and has been found to recruit eosinophils in a guinea pig model [211,212]. The relationship between LTB4 and airway remodeling has not been fully studied, but BLT1 and BLT2 are expressed on HASM cells. LTB4 has been shown to induce HASM cell migration and proliferation in vitro [213]. Therefore, LTB4 might be involved in airway remodeling in asthma. 

### 6.2. IL-8

IL-8 is increased in sputum or BALF from patients with severe asthma and is inversely correlated with %predicted FEV1 and sputum neutrophil counts [23,24,59,214,215,216]. A recent study showed that IL-8 in BALF was the only cytokine that distinguished controlled from uncontrolled asthma among 48 evaluated cytokines [216]. IL-8 has been shown to induce HASM cell proliferation and migration [217,218,219], to stimulate mucin secretion [220], and to upregulate MUC5A and MUC5B in goblet cells [221]. YKL-40 has been shown to induce IL-8 in bronchial epithelial cells and to cause HASM cell proliferation and migration [222]. Therefore, IL-8 might be related to severe neutrophilic asthma and airway remodeling in asthma. 

### 6.3. TNF-α 

TNF-α is a proinflammatory cytokine related to neutrophilic asthma. In vitro, TNF-α induced airway smooth muscle migration and proliferation [223], extracellular matrix deposition, subepithelial fibrosis, and inflammatory cytokine secretion [224,225]. In a mouse model, TNF-α was involved in glucocorticoid insensitivity in neutrophilic inflammation in asthma, which may induce chronic inflammation and lead to airway remodeling [226]. In vitro, miR874, which may be associated with the development of asthma, has been shown to inhibit TNF-α-induced IL-6, IL-8, collagen I, and collagen III production in ASM cells [224].

### 6.4. IL-17A

IL-17A is an independent risk factor for severe asthma and is involved in obesity-associated asthma and CS-related airway neutrophilia [82,163,227]. In a mouse model, IL-17A induced type V collagen expression, *TGFB1* mRNA expression, and SMAD3 activation in airway epithelial cells [228]. In vitro, MUC5A and MUC5B expressions have been induced by IL-17A via IL-6 and NF-κB in epithelial cells [229,230,231]. IL-17A has also been shown to be involved in the epithelia mesenchymal transition via expression of TGF-β1 in airway epithelial cells [232]. In a mouse model, IL-17 was involved in airway smooth muscle hyperplasia mediated by fibroblast growth factor 2 from airway epithelial cells, and neutrophil elastase played an important role in this model [233,234]. In other mouse models using OVA and LPS for exacerbation, anti-IL-17A antibody decreased extracellular matrix deposition [235] and vascular remodeling [234]. Therefore, IL-17A comodulated with TGF-β1 is involved in airway remodeling in asthma and is related to neutrophils [236].

### 6.5. Other Inflammatory Mediators and Cytokines

IL-1β has been shown to induce neutrophilic asthma and IL-33 expression in a mouse model of asthma with viral infection exacerbation [148], and was a key cytokine in induction of airway smooth muscle hypersensitivity [237]. IL-1β alone or with TNF superfamily members has been observed to cause airway neutrophilic inflammation and remodeling in an adult animal model [238,239]. Oncostatin M (OSM) is released from neutrophils and induces epithelial barrier dysfunction [240]. In severe asthma, there are increases in OSM in sputum and in OSM-positive neutrophils in biopsy specimens [241]. OSM is also increased in patients with asthma with fixed airway obstruction [242]. Furthermore, MMP9 and elastase may be involved in airway remodeling in asthma [243,244,245].

## 7. Treatment

Treatment with an ICS is a key approach for asthma, but corticosteroids are not effective in neutrophilic asthma [246,247]. Treatment of asthma related to neutrophilic inflammation can be categorized into non-pharmacological approaches, nonspecific treatment for neutrophil inflammation, treatment specific to neutrophils and neutrophil mediators, and biologics (Table 2). 

### 7.1. Non-Pharmacological Approach

Smoking cessation may be the best way to reduce neutrophilic inflammation in neutrophilic asthma patients who smoke. In a clinical trial, smoking cessation in young adults with asthma improved asthma control, but with persistent eosinophil counts and little neutrophil reduction [248]. In this trial, 17% of the subjects had neutrophilic asthma. Another clinical trial showed improvements in lung function and sputum neutrophil counts [151]. Weight loss by diet, exercise, diet with exercise, or surgical intervention also improved asthma control, quality of life, lung function, and airway hyper-responsiveness [249,263,264,265,266]. Thus, smoking cessation and weight loss are good approaches for patients with severe asthma, regardless of the inflammatory phenotype. 

### 7.2. Nonspecific Treatment for Neutrophilic Inflammation 

#### 7.2.1. Macrolides

Macrolides have various functions, in addition to their actions as antibiotics [267]. The effectiveness of clarithromycin has been shown in chronic stable asthma with *Mycoplasma pneumoniae* or *Chlamydia pneumoniae* mRNA in the airway [268]. The AMAZES study showed the effectiveness of azithromycin for persistent uncontrolled asthma [269]. In this study, 43% of the cases were eosinophilic, 11% neutrophilic, 30% paucigranulocytic, and 4% mixed, based on sputum phenotyping. A subset analysis in the AMAZES study showed that azithromycin was similarly effective for severe asthma in the cases with an eosinophilic sputum phenotype [269]. The effect of azithromycin was correlated with the abundance of *Haemophiles influenzae* colonization as assessed by quantitative polymerase chain reaction [270]. In the AMAZES study, sputum TNFR1 and TNFR2 were increased in neutrophilic asthma and azithromycin reduced sputum TNFR2 in non-eosinophilic asthma, which may be related to the therapeutic mechanism [271]. The AZISAST study showed a reduced rate of severe exacerbation by azithromycin in non-eosinophilic severe asthma [272]; in a study in severe neutrophilic asthma, 8-week administration of this drug improved quality of life and reduced airway IL-8 and neutrophils [250]. Therefore, long term macrolide treatment is a promising therapy in severe asthma, particularly for the neutrophil-dominant phenotype.

#### 7.2.2. Phosphodiesterase Inhibitors

Roflumilast is an oral phosphodiesterase (PDE) inhibitor that has therapeutic effects on COPD [273] and asthma-COPD overlap [274]. Several studies have shown the efficacy of roflumilast alone [275,276] or in combination with a leukotriene receptor antagonist in moderate-to-severe asthma [277]. Roflumilast attenuates both eosinophilic and neutrophilic inflammation induced by allergens [251,278]. Inhaled PDE inhibitors have also been examined in patients with asthma (reviewed in [279]): CH6001 showed inhibition of the late asthmatic response induced by allergen exposure [280] and RPL554 (a PDE3 and PDE4 inhibitor) increased FEV1 in patients with asthma and reduced neutrophils and total cells in sputum from healthy individuals [281]. Studies of PDE inhibitors focusing on neutrophilic asthma are needed, but roflumilast and inhaled PDE4 inhibitors may be promising for neutrophilic asthma [282].

#### 7.2.3. Anticholinergics

Anticholinergics have been used for treatment of COPD and asthma. Long-acting muscarinic antagonist (LAMAs) and short-acting muscarinic antagonists are both available for treatment of asthma. LAMAs decreased eosinophils in sensitized mice [283,284], and in an obstructive airway disease model in rat, tiotropium decreased neutrophil counts, IL-1β and IL-6 in bronchoalveolar lavage [285]. In an in vitro study in human epithelial cells, tiotropium reduced IL-8 production induced by IL-17A [286] or LPS [287]. In clinical studies, tiotropium has been shown to be effective as an add-on therapy to ICS [288] or ICS/LABA [289] in uncontrolled asthma, and Iwamoto et al. found that anti-cholinergics were effective in non-eosinophilic asthma [290]. Tiotropium has been shown to be effective, independent of type 2 inflammation in adults [252,291,292] and in children and adolescents [253]. However, the efficacy of ICS or tiotropium was similar to that of a placebo in patients with mild persistent asthma, including 73% with low eosinophilic asthma [293]. 

### 7.3. Specific Therapy for Neutrophils and Neutrophil Mediators

#### 7.3.1. CXCR2 Antagonists

CXCR2 is a receptor for IL-8 that is expressed on neutrophils. A CXCR2 inhibitor, SCH527123, reduced sputum neutrophils and exacerbation in severe asthma cases in a 4-week clinical trial [254]. Another CXCR2 antagonist, AZD5069, reduced neutrophils in bronchial mucosa, sputum, and blood, but failed to reduce severe exacerbation [294,295]. 

##### 7.3.2. 5-Lipoxygenase-Activating Protein Inhibitors and 5-Lipoxygenase Inhibitors 

Five-lipoxygenase-activating protein (FLAP) and 5-lipoxygenase (5-LO) are required for synthesis of LTB4. GSK2190915 is a FLAP inhibitor that has been evaluated for patients with asthma in several studies [255,296,297]. In one study focused on neutrophilic asthma, a FLAP inhibitor suppressed sputum LTB4 and urine LTE4 levels, but failed to reduce neutrophil counts in sputum and had no clinical effects on FEV1, PEF, and ACQ scores [296]. Zileuton is a 5-LO inhibitor that has also been evaluated in patients with asthma [256,298] and has been shown to be effective in moderate-to-severe asthma based on improved PEF and asthma symptoms [256]. A recent retrospective study showed no associations among Th2-high or Th2-low phenotypes and a poor response rate to zileuton in association with severe asthma and obesity [298].

### 7.4. Biologics

Several biological agents are currently available for patients with severe asthma. There are six FDA-approved monoclonal antibodies (mAbs): omalizumab, which is anti-IgE antibody; mepolizumab and reslizumab, which are anti-IL-5 antibodies; benralizumab, which is an anti-IL-5 receptor α antibody; dupilumab, which is an anti-IL-4 receptor α antibody; and tezepelumab, which is an anti-TSLP antibody. These biologics exhibited clinical benefits for allergic/Th2-high asthma [299].

#### 7.4.1. Targeting TSLP

Tezepelumab, a humanized mAb for TSLP, has been tested in a phase 2 clinical trial in patients with moderate-to-severe asthma [300] and in a phase 3 clinical trial in patients with severe asthma [257]. Tezepelumab reduced the rate of exacerbation and improved FEV1, ACQ, and AQLQ scores, regardless of type 2 inflammation. Therefore, tezepelumab may be effective for severe neutrophilic asthma. Biphasic antibodies for TSLP/IL-13 (zweimab and doppelmab) have recently been developed [301] and may also be evaluated for treatment of severe asthma with type 2, non-type 2, or neutrophilic inflammation. 

#### 7.4.2. Targeting TNF-α

Blocking of TNF-α by infliximab and golimumab, which are anti-TNF-α mAbs, and etanercept, which is a recombinant TNF-α receptor, has been examined as treatment for moderate and severe asthma [258,259,302,303,304]. In patients with severe and uncontrolled asthma under treatment with high-dose ICS and LABAs, golimumab did not improve FEV1 or the rate of exacerbation [258]. Etanercept, in several clinical trails, has been shown to improve airway hyper-responsiveness (AHR); FEV1, AQLQ, and ACQ scores; and asthma symptoms; as well as to reduce sputum macrophages and CRP levels in several clinicals trials [302,303,304]. However, a large, randomized clinical study of etanercept for moderate-to-severe asthma showed no efficacy for ACQ, AQLQ, FEV1, exacerbation rate, or AHR [259]. 

#### 7.4.3. Targeting IL-17

Anti-IL-17 antibody has been shown to decrease airway hyper-responsiveness and airway inflammation in a mouse model of obesity, alone [305] or in combination with a Rho-kinase inhibitor [306]. Secukinumab, an mAb targeting IL-17A, was tested in a randomized clinical trial in patients with severe asthma treated with high doses of ICS alone or in combination with a LABA. In this trial, responders (defined as patients with a 5% increase in predicted FEV1) showed increased nasal epithelial neutrophilic inflammation and had decreased markers of IgE-driven systemic inflammation based on a nasal brushing pathway analysis of differentially regulated genes [307]. A randomized, double-blind, placebo-controlled study of brodalumab, a monoclonal antibody targeting IL-17 receptor A, showed no treatment effect in subjects with moderate-to-severe asthma [260]. A bispecific antibody targeting IL-13 and IL-17 showed clinical safety with no deaths or serious adverse events in a phase I study [308].

#### 7.4.4. Targeting IL-23

As mentioned above, IL-17 is involved in neutrophilic inflammation in asthma and IL-23, an IL-12 family cytokine, is important for maintenance and recruitment of Th17 cells [309]. However, risankizumab, an IL23p19 mAb, failed to show efficacy for worsening of asthma as compared with a placebo in a phase I, randomized, double-blind, placebo-controlled study in adults with severe asthma, with no significant changes in sputum cell differentials [261]. 

#### 7.4.5. Targeting IL-6

Tocilizumab, an anti-IL-6 receptor mAb, had effects on CRP, IL-6, and soluble IL-6 receptor, but did not improve allergen-induced bronchoconstriction in 11 patients with mild asthma [262]. 

### 7.5. Other Potential Therapy for Neutrophilic Asthma

Peroxisome proliferator-activated receptor-gamma agonists have been tested in a murine model of neutrophilic asthma [310]. Statins are also candidate drugs for patients with obesity and asthma [311,312]. Inhibitors of protein kinases, p38 MAPK, and phosphoinositide 3-kinase (PI3K δ and γ) have been examined for COPD or asthma [312,313,314,315]. These inhibitors might be effective in neutrophilic asthma because the PI3K pathway is involved in neutrophil migration and degranulation [316,317]. Glucagon-like peptide-1 receptor (GLP-1R) agonists inhibit aeroallergen-induced activation of ILC2 and neutrophilic airway inflammation in obese mice [318]. Fore et al. found that patients with asthma who received GLP-1R agonists had less exacerbation than those treated with sulfonylureas or insulin [319]. Some of these drugs have been tested for asthma or COPD, but not specifically for neutrophilic asthma. 

## 8. Conclusions

Asthma is a heterogenous syndrome that includes neutrophilic asthma as one phenotype. There is still uncertainty about this phenotype, but many studies have shown the importance of neutrophils in asthma. There is no clear definition of neutrophilic asthma, but sputum and peripheral blood neutrophils, YKL-40, H_2_S, MPA, and miRNAs may be useful biomarkers for this condition. Identification of new biomarkers or combinations of biomarkers will be important for future diagnosis of neutrophilic asthma. Neutrophilic inflammation is involved in airway remodeling in patients with asthma, including those with obesity and GERD. Non-pharmacological and pharmacological therapy, including targeting of neutrophils and nonspecific treatment, may be useful for neutrophilic asthma, but most treatments have yet to be tested in patients with this condition. Further studies, focused on non-type 2 cases and neutrophilic inflammation, are needed to develop treatment for severe neutrophilic asthma. 

## Figures and Tables

**Figure 1 diagnostics-12-01175-f001:**
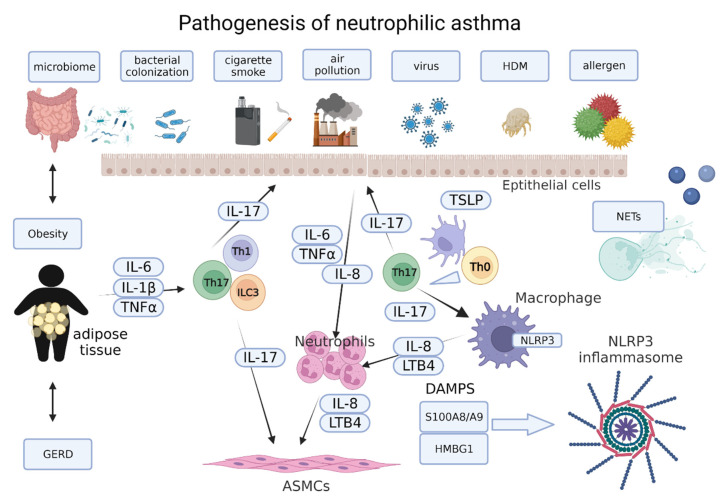
Pathogenesis of neutrophilic asthma. Several cells, including airway epithelial cells, macrophages, T helper (Th) cells, innate helper 3 cells (ILC3), airway smooth muscle cells (ASMCs), and neutrophils play important roles in the pathogenesis of neutrophilic asthma. Airway epithelial cells, stimulated by air pollution, cigarette smoke, bacterial colonization, virus, and allergens, secrete TSLP, IL-33, and IL-25. TSLP secreted from epithelial cells and inflammatory cells converts Th0 to Th17 cells and subsequently induced neutrophil recruitment via IL-8 and GM-CSF, induced by IL-17 from airway epithelial cells. The IL-17/Th17 axis is involved in bacterial colonization and microbiome associated neutrophilic inflammation in asthma. Obesity and GERD are related to severe, neutrophilic asthma and the IL-17/Th17 axis is involved in these conditions. Neutrophil extracellular trap (NETs) formation, damage-associated molecular patterns (DAMPs), and NLPR3 inflammasome are also involved in the pathogenesis of neutrophil asthma.

**Figure 2 diagnostics-12-01175-f002:**
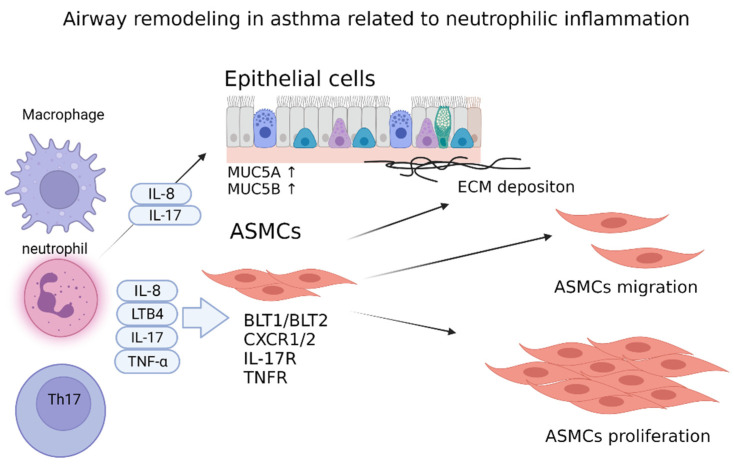
Airway remodeling in asthma related to neutrophilic inflammation. Airway remodeling in asthma is a characteristic feature of chronic asthma. LTB4, IL-8, LTB4, and TNF-α are elevated in an asthmatic airway and are related to airway remodeling. LTB4, IL-8, and TNF-α induce airway smooth muscle cell proliferation and migration. IL-8 and IL-17 upregulate MUC5A and MUC5B expression in epithelial cells. Abbreviations: IL, interleukin; LTB4, leukotriene B4, TNF-α; Tumor necrosis factor α, BLT1/2: leukotriene B4 receptor 1/2, IL-17R: IL-17 receptor, TNFR: TNF receptor, ASMCs: airway smooth muscle cells.

**Table 1 diagnostics-12-01175-t001:** Possible biomarkers for neutrophilic asthma.

Biomarker	Sample	Definition	Significance	Refs.
**YKL-40**	Serum, sputum	Not established, but serum YKL-40 > 60.94 ng/mL showed impaired lung function and require corticosteroid	YKL-40 is released from neutrophil and epithelial cells, YKL-40 is released from neutrophils and epithelial cells Serum YKL-40 correlates with sputum neutrophil counts	[181,182,183]
**Hydrogen sulfide (H_2_S)**	Serum, exhaled breath, sputum	Not established	Sputum H_2_S correlates with the degree of airflow limitation Serum/sputum H_2_S predicts asthma exacerbation	[184,185,186]
**MPO**	Sputum	Not established	Sputum MPO correlates with sputum YKL-40 and neutrophils	[23,187]
**Neutrophil**	Serum, sputum	Sputum > 60% or 76%	Associated with chronic airway obstruction, annual decline of FEV1	[188,189]
**MicroRNA**	Sputum, serum, and plasma	Not established	miR-199a-5p, miR142-3p, miR233-3p, and miR629-3p are increased in neutrophilic asthmamiR299a -5p is negatively correlated with FEV1	[190,191]

**Table 2 diagnostics-12-01175-t002:** Summary of treatment for asthma related to neutrophilic inflammation.

Non-Pharmacological Approach			
Approach	Patient Population	Outcomes	Ref.
Smoking cessation	Young patients with asthma (19–40 years old), steroid-free, 17% neutrophilic asthma	Improved asthma control and flung function	[248]
Weight loss	18–75-year-old, obese patients with asthma (BMI > 35 kg/m^2^)	Improved asthma control, QOL, lung function, and AHR	[249]
Nonspecific treatment for neutrophilic asthma			
Therapy	Patient population	Outcomes	Ref.
Macrolide (azithromycin, clarithromycin)	Non-eosinophilic or neutrophilic severe asthma (18–75-year-old patients)	Reduced asthma exacerbation, QOL, and lung function	[250]
PDE inhibitor	Patients 18–70 years of age, moderate-to-severe asthma	Improved lung function and asthma control	[251]
Tiotropium	Adult symptomatic patients with asthma despite treatment with medium-dose ICS	Improved lung function and asthma control, reduced risk of severe exacerbation, independent of type 2 inflammation	[252]
Tiotropium	6–17-year-old patients, symptomatic severe asthma	Improved lung function and ACQ, reduced risk of exacerbation, independent of type 2 inflammation	[253]
Specific treatment for neutrophil and mediators			
SCH527123/CXCR2	Severe asthma and sputum neutrophil >40%	Fewer mild exacerbations and a trend towards improvement in the ACQ, but not statistically significant	[254]
GSK2090915/FLAP	Persistent asthma treated with SABA only	Improved symptom score and reduced SABA use	[255]
Zileuton/5-LO	Moderate-to-severe asthma treated with low dose ICS	Improved PEF and symptoms	[256]
Biologics			
Tezepelumab/TSLP	Moderate-to-severe asthma	Reduced rate of exacerbation, improved lung function, ACQ, and AQLQ, regardless of type 2 inflammation	[257]
Golimumab/TNF-α	Uncontrolled asthma with high-dose ICS/LABA	No improvement in FEV1 and exacerbation	[258]
Etanercept/TNF-α	Moderate-to-severe persistent asthma	No improvement in FEV1 and ACQ, exacerbation, AHR, AQLQ	[259]
Brodalumab/IL-17 receptor	Inadequately controlled moderate-to-severe asthma treated with high-dose ICS ± LABA	No treatment differences were observed	[260]
Risankinumab/IL-23	Adult patients with severe asthma	No improvement in asthma exacerbation	[261]
Tocilizumab/IL-6	Mild asthma	No improvement in allergen-induced bronchoconstriction	[262]

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
