# Peer review of "Neutrophils and Asthma"

_diagnostics, 2022, doi:10.3390/diagnostics12051175_

Round 1

Reviewer 1 Report

From my point of view the article reviews a topic that has several knowledge gaps in the literature that you properly emphasize. Before being suitable for publication I just suggest a language review.

Author Response

Thank you for your comments. In accordance with the suggestion from the previous comment, we understand that many review articles on this topic have been published recently. Accordingly, we have cited recently published papers. We have also revised the description in page 3, lines 62 to 67, in page 4, lines 165 to 167, in page 7, lines 286 to 288, in page 10, lines 396 to 397, and in page 15, lines 540 to 541. These changes are highlighted in yellow. We have modified the title of figure 2 and description about receptors for airway smooth muscles. We have proofread this paper again. 

Reviewer 2 Report

My comments have been addressed well. I have no further reservations.

Author Response

Thank  you for your comments. We have proofread this paper again.

Reviewer 3 Report

This is a revised version of the manuscript. 

Authors have comprehensively discussed neutrophils and asthma, from definitions to therapeutics. The authors have diligently covered most of the aspects of neutrophil asthma, including epidemiology, causative agents, pathophysiology, biomarkers, and therapeutics. 

The authors have addressed the worthy reviewer's comments and improved the manuscript meaningfully. 

Overall the revised manuscript is acceptable for publication in its current form. 

Author Response

Thank you for your comments.

This manuscript is a resubmission of an earlier submission. The following is a list of the peer review reports and author responses from that submission.

Round 1

Reviewer 1 Report

Dear authors,

Thank you for your submission and work. I think the article is considerably well written and interesting but I have issues but it's originality and interest. There are more than one review article about this topic in the last year, namely: https://thorax.bmj.com/content/76/8/835.long

https://pubmed.ncbi.nlm.nih.gov/34547115/

I would recommend you to rewrite/resubmit another paper just focusing in a certain particular aspect of neutrophilic inflammation where you paper really is novel.

Reviewer 2 Report

With interest, I read the manuscript diagnostics-1416005.

In this manuscript, the Authors demonstrate a deep knowledge oft he topic and/or invested lots of time in the preparation of this work. In any case, impressive.

Specific comments:

1. This manuscript would benefit much if it was additionally equipped with 5-6 comprehensive figures illustriating ist content, or at least ist most important parts. If the Authors find it better, some aspects could be summarized in the tables instead of illustrating them with the figures. Nevertheless, substantially expanded graphical represenation oft he manuscript content is more than necessary to make it more “digestible“ to the Reader.

2. The two tables already present in the manuscript need to be improved.

2A. Table 1 is graphically poor in general. Also some minor aspects need to be corrected, suhc as “correlate“ or “correlates“? Starting a new cell with upper case or not (in any case, always the smae way). Etc.

2B. Table 2 is very ascetic, noth graphically and, especially, regarding ist content. Both aspects should be elaborated.

3. In the Abstract (e.g. “NETs“), but also in the main text (e.g. “Th2“), some abbreviations are inntroducte that are not explained. Please, correct.

4. Besides, please, make sure the the text is uniform, e.g. that the abbreaviations in the asbtract are the same as in the main text ((e.g. “NETs“ vs. “NETS“).

5. Line 132. Whe you write “leptins“, I guess, you mean adipokines. Please, refer to PMID: 30057383 and re-write a bit by expanding on different adipokines (2-3 sentences).

6. From line 504. In this chapter in turn, please, refer also to PMID: 33926084.

7. Neutrophils in asthma are interconnected with other cell types also using exosomes/extracellular vescicles (PMID: 34067156). Please, mention shortly.

Reviewer 3 Report

Yamasaki et al., have submitted the review entitled “Neutrophils and asthma”.

The authors have covered a broad area of the research focusing on the contribution of neutrophils in asthma. The authors didactically present the introduction with types of asthma, pathogenesis, list of biomarkers, and therapeutic avenues.

  • Although the review is comprehensive, it lacks representative figures. Therefore I request the authors to provide the illustrations, which represents the majority of the text. “The more illustrations-the more visibility.”
  • In addition, I would like to suggest the authors to provide the list of salient features and side effects of each treatment in a table format.